# Feasibility Study on Earthquake Prediction Based on Impending Geomagnetic Anomalies

**Ying Huang** [1], **Peimin Zhu** [2,*] and **Shaodong Li** [1]

1. Institute of Geography and Information Engineering, China University of Geosciences, Wuhan 430074, China; hying@cug.edu.cn (Y.H.)
2. School of Geophysics and Geomatics, China University of Geosciences, Wuhan 430074, China
* Correspondence: zhupm@cug.edu.cn; Tel.: +86-139-7116-6826

**Abstract:** By deploying a magnetic monitoring network in the earthquake-prone areas of Sichuan, China, and conducting long-term observations, processing, and analysis of real-time geomagnetic data, it can be observed that the pre-earthquake geomagnetic anomalies are highly correlated with the occurrence time of earthquakes. We propose a novel algorithm that obtains a new quantity by accumulating geomagnetic anomaly energy to eliminate external environmental interference and take its gradient as a measure for predicting the occurrence time of an earthquake. Through observations of a large amount of geomagnetic data, it is confirmed that the proposed method can be used to predict the occurrence time of earthquakes with about 75% to 85% accuracy. Conclusions: The geomagnetic anomaly phenomenon can be accurately observed and recorded before an impending earthquake, and it has been confirmed by data that using this observation makes imminent earthquake prediction a practical prediction method.

**Keywords:** magneto seismic effect; eearthquake prediction; magnetic data monitoring; pre-earthquake geomagnetic anomaly; geomagnetic anomaly energy

## 1. Introduction

Many observations show that strong geomagnetic anomalies occur repeatedly before large earthquakes. On 27 March 1964, an earthquake of M9.2 occurred in Alaska, the United States, and at 440 km away from the epicenter 66 min before the earthquake, a magnetometer recorded an extraordinary magnetic anomaly of 100 nT [1]. This geomagnetic anomaly also began to be reported before earthquakes in the late 1980s. The abnormal changes are observed in the super low frequency (ULF) range. On 17 October 1989, an M7.1 magnitude earthquake occurred in Loma Prieta, California. On the day of the earthquake, the geomagnetic ULF intensity was 20 times stronger than usual, and the signal increased by 60 times three hours before the earthquake [2]. During 1–15 July 2000, the Izu Island volcano in Japan experienced three earthquakes greater than M6.0 (M 6.4/6.1/6.3), with significant geomagnetic anomalies occurring 69 days to 40 min before the earthquake [3]. For the M8.0 Wenchuan earthquake in China on 12 May 2008, research shows that the geomagnetic field had significant changes from 1.5 days to several hours and several minutes before the earthquake [4]. The maximum changes in each magnetic component recorded by the Chengdu geomagnetic station were magnetic declination $\Delta D \approx 26S$, horizontal component $\Delta H \approx 28S$, vertical component $\Delta Z \approx 95S$ and total field strength $\Delta F \approx 31S$, of which S represents standard deviation of the local geomagnetic field at normal times. Before the "311" earthquake with a magnitude of 9.0 (11 March 2011) and the "411" earthquake with a magnitude of 7.3 (11 April 2011) in the Eastern Sea Area of Japan, there were obvious geomagnetic anomalies [5]. The latest article published by Li Junhui et al. in the journal "Earthquake" in January 2022 described that before the 6.6 magnitude earthquake in Min County and Zhang County, China on 22 July 2013, the third principal component with

the smallest geomagnetism showed a significant enhancement change from 20 April to 10 June 2013, exceeding the threshold, and the anomaly continued for about 50 days [6] A typical example is the VAN method (VAN is the abbreviation of three Greek scientists, P. Varotsos, K. Alexopoulos and K. Nomicos) [7] This technology was been developed in Greece in the early 1980s and has also been applied in Japan since the 1990s. The seismic electrical signal in the VAN method is the transient DC ground potential change observed before earthquakes by burying the dipole of the electrode. From both the experimental and theoretical perspectives, the VAN method is by far the best-equipped method in this category. However, in modern society, artificial magnetic interference from electrical equipment seriously affects the accuracy of the VAN method. In addition, its prediction time exceeds one week, reaching several weeks, which is still too long to be practical [8].

The mechanism of geomagnetic anomalies before and after an earthquake is still in being researched. However, some articles have verified using experimental observation that squeezing, friction, impact and the accompanying rock fracturing process will produce electromagnetic anomalies, with frequencies ranging from 0 Hz to 10 kHz [5]. Before the earthquake, compression, friction, impact and rock fracturing processes will be formed, which can explain the rationality of geomagnetic anomalies before and after the earthquake.

From the cases observed in the above literature, it can be seen that they are all cases of geomagnetic anomalies observed before large earthquakes, and there are no cases of small earthquakes. Most of the articles describe only one observation of geomagnetic anomalies before an earthquake, rather than consecutive occurrences, and most of the literature is post-earthquake data analysis. These articles can only explain that the geomagnetic anomaly caused by the earthquake is occasionally observed, or the data are found inversely after the earthquake. But can it be observed before every earthquake? This paper, through a large number of observations of anomalies using the intensive observation method, proposes an independent and innovative algorithm to capture the data for geomagnetic anomalies before earthquakes. It shows that the geomagnetic anomalies caused by earthquakes can be observed before earthquakes with high probability, and can be used for earthquake prediction.

## 2. Methods for Observing and Capturing Geomagnetic Anomalies

What is geomagnetic anomaly? The Earth's magnetic field is usually regarded as a stochastic process $f(t)$ with a mean value of a certain value, and the variation range in its normal value is usually not more than tens of $nT$ up and down in the mean value. In theory, when the difference between the value measured by a magnetic intensity instrument at a certain time and the mean exceeds 3–5 times its mean square deviation, it is considered that its value is abnormal. When this magnetic anomaly is caused by underground geological changes, it is called a geomagnetic anomaly.

Assume that a discrete time series of geomagnetic observation, $f(k)$, as an ergodic stationary stochastic process. Its mean can be expressed as follows:

$$\mu_f = E\{f(k)\}, \tag{1}$$

and its standard deviation is $S = \sqrt{\frac{1}{N}\sum_{k=1}^{N}\left[f(k) - \mu_f\right]^2}$. Geomagnetic anomaly $\Delta F$ is defined as:

$$\Delta F = \left[f(t) - \mu_f(t)\right] \geq (3 \sim 5)S, \tag{2}$$

A relatively dense magnetic sensing network was deployed in Liangshan Prefecture, Sichuan Province, China, which was prone to earthquakes in 2019. The magnetic intensity sensors are located in an area that is 88 km long and 154 km wide, with a total of about 30 portable three-component magnetometers of model AM2-1001 with a sensitivity of 1 nT, and its performance parameters are shown in Table 1. The stations are distributed in the range of longitude 101.71°~102.62° E and latitude 27.29°~28.92° N. The distribution of the stations is based on the local terrain and residential areas, and the average distance between

adjacent magnetic sensors is about 20 km. Each magnetometer records the three components x, y, and z of the geomagnetic field at its position every 3 s, and transmits the recorded data back to the server through the China Mobile 4G network. Since September 2019, the monitoring has lasted for approximately four years, with approximately 100,000 bytes of data per day. As of now, approximately 1.25 GB of data have been obtained.

**Table 1.** Specifications of AM2-1001 sensor.

| Meas. Range | ±100 µT, Other Ranges on Request |
|---|---|
| Accuracy at 20 °C | ±2% ± 0.3 µT |
| Operating temperature | −40 to +85 °C |
| Zero drift | <2 nT/K |
| Output voltage OUT+ ref. to OUT− | ±1 V/50 µT, max. ±2.5 V |
| Bandwidth | 0 to 1 kHz (−3 dB) |
| DC output impedance | <1 Ω |
| Reference output OUT− | 2.5 V ref. to supply ground (0 V) |
| Max. load between OUT+ and OUT− | >1 kΩ, <100 pF |
| Noise | <0.5 $nT_{RMS}$ or 3 $nT_{PP}$ (0.1 to 10 Hz), ~150 pT/$\sqrt{Hz}$@ 1 Hz |
| Supply voltage | 5 V ± 5% |
| Supply current | ~2 mA |
| Dimensions | 44.5 mm × 14 mm × 5.5 mm |
| Length of detection coil | 22 mm |

### 2.1. Geomagnetic Anomaly under Intensive Observation

In the densely arranged magnetic strength sensor network, when the underground geology changes, it can be observed that many stations will have magnetic anomalies at the same time. The situation is the same as shown in Figure 1; the points with red circles have magnetic anomalies. Moreover, the magnetic anomaly with synchronous change characteristics is considered to be the geomagnetic anomaly caused by earthquakes. Figure 2 shows the waveform of a geomagnetic anomaly with synchronous change characteristics.

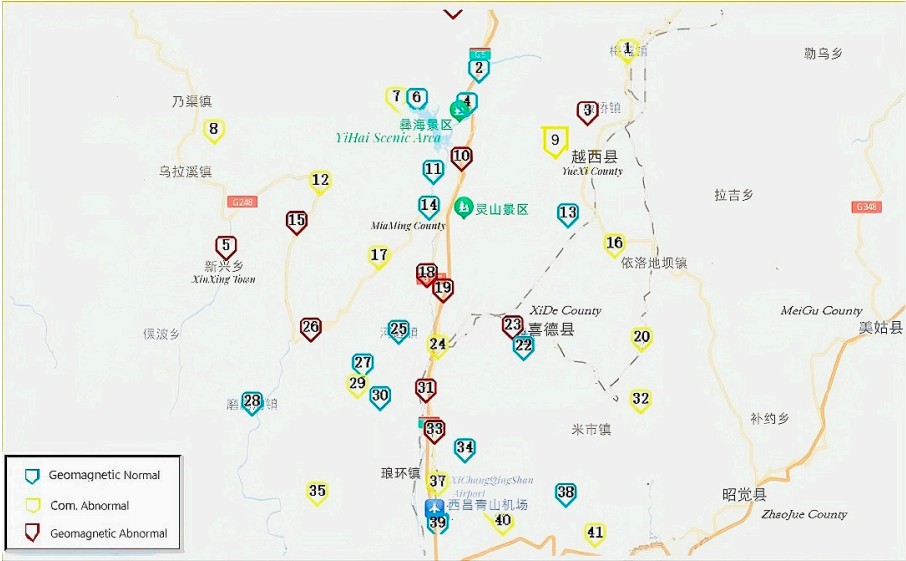

**Figure 1.** Magnetic anomaly in magnetic intensity sensor network (the numbers indicate the station serial numbers).

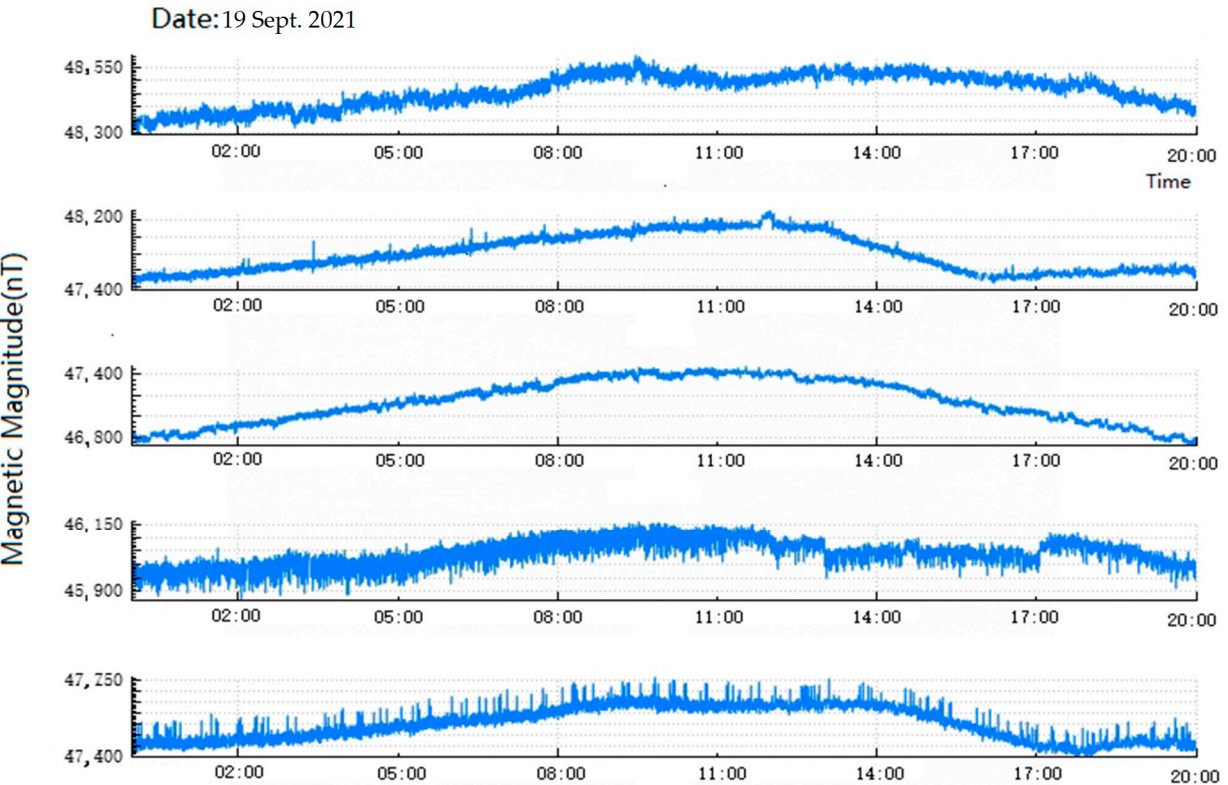

**Figure 2.** Magnetic magnitude waveform and synchronous enhancement before earthquakes.

### 2.2. Extraction of Geomagnetic Anomaly Signals Caused by Earthquakes

Due to the complex environment of modern human life with various electrical equipment and stations, there are various electromagnetic interferences around monitoring points. The magnetic intensity sensor in the magnetic sensor network in this experiment does not use any shielding interference device, so it will be subject to various interferences. Generally, the geomagnetic anomaly of small earthquakes is very weak, and may be submerged in environmental noise. In order to extract the geomagnetic anomaly in this environment, it is necessary to filter out all kinds of magnetic anomalies or background noise not caused by earthquakes. The magnetic interference around the observation environment mainly includes the following categories:

(1) Random interference: surrounding electrical appliances, vehicles, and lightning. This interference only causes short-term interference to individual stations.

(2) Static interference: high-voltage stations, communication base stations, underground metal pipelines, etc. in the environment, which only cause long-term interference to individual stations.

(3) The annual and daily variations from the solar system and the Earth's magnetic field can be corrected by the geomagnetic correction method.

The idea of our algorithm for geomagnetic anomaly capturing is as follows:

(1) Interpretation of anomaly: using the window-based magnetic anomaly judgment method, when the average number in a period exceeds a certain threshold, it is considered as the abnormal magnetic field strength, referred to as a geomagnetic anomaly in short. This algorithm can eliminate short-term spike anomalies, such as magnetic anomalies caused by automobile passing and electrical switches.

(2) Similarity judgment: ignoring the anomalies of individual stations, which is recognized as a geomagnetic anomaly when the data from multiple stations have similar changes during the same time period which may last for a few days.

(3) Using the energy integration method to improve the signal-to-noise ratio is as follows:

In this algorithm, the magnetic intensity signal is considered a stationary random process, and its mean is obtained by calculating the cumulative mean, which is the arithmetic mean $\mu(0)$ of all the magnetic intensity sample values in one day, and then it is averaged using the previous day's mean to obtain the daily average $\mu(i)$ as the following:

$$\mu(i) = \frac{1}{2}[\mu(i-1) + \mu(0)] \tag{3}$$

The daily observation time is divided into many observation windows, for example, $N$, where the window length is $T$. We newly define $F$ as the total geomagnetic field after removing the variation of daily and annual magnetic field, and $\Delta F$ as the anomalous intensity of the net geomagnetic field only caused by earthquakes or other geological factors, for example, the iron ores. Especially, within a time window of length $T$, we define $E = \Delta F^2 \times T$ as a new measure of the geomagnetic field observation, i.e., the cumulative geomagnetic anomaly energy, which can enhance the pre-earthquake geomagnetic anomaly signal, eliminate random interference from the surrounding environment, and highlight weak pre-earthquake geomagnetic anomaly change signals. We prove the above conclusion as follows:

First, we assume that within kth time window the geomagnetic anomaly energy at discrete time $k$ is $E(k)$, the noise energy is $N(k)$. The signal-to-noise ratio in this time window can be expressed as follows:

$$SNR = \frac{E(k)}{N(k)} = \frac{\Delta F^2 \times T}{\mu_k^2 \times T} \tag{4}$$

where $\mu_k$ is the mean of the random noise at time $k$. When the geomagnetic field is stable, the fluctuation in the magnetic field is usually less than 50 nT, that is $\Delta F < 50$ nT, $E(k)$ tends to stabilize and can be approximated as a constant.

When the fluctuation in the geomagnetic field increases, $\Delta F$ shows a gradual increase or decrease, and $\Delta F^2 \times T$ follows changes.

In one whole day, the total signal-to-noise ratio of the accumulated geomagnetic anomaly energy obtained by cumulative summation:

$$SNR = \sum_{k=1}^{N} \frac{E(k)}{N(k)} \tag{5}$$

The stronger the $\Delta F$, the significantly stronger the *SNR*. Therefore, observing the cumulative geomagnetic energy can effectively eliminate the surrounding magnetic interference and capture the weak geomagnetic anomalies before earthquakes.

Our new algorithm for capturing impending geomagnetic anomalies is depicted as follows. Assume that geomagnetic field anomalies occur a time window $(t', t' + T)$ set at the beginning of a certain time $t'$. Calculate the cumulative geomagnetic anomaly energy $E(t', t' + T)$ according to the below formula:

$$E(t', t' + T) = \sum_{m=1}^{M} \int_{0}^{T} \Delta F^2(t)dt \tag{6}$$

where $M$ indicates the number of geomagnetic monitoring stations. So, we can derive the geomagnetic anomaly gradient (Equation (7)) for the earthquake prediction analysis.

$$G = \frac{1}{M} \frac{d}{dt} E(t', t' + T) \tag{7}$$

### 2.3. Research on Rules of Earthquake Prediction

The above algorithm is extended into daily magnetic field monitoring, and the cumulative geomagnetic anomaly energy and geomagnetic anomaly gradient are calculated once a day. The multi-day gradient *G* forms a time sequence *A*. If a threshold is set and

an extraction algorithm is applied to sequence $A$, a prediction time sequence $B$ can be extracted according to the following rules which we propose:

(1) The daily geomagnetic anomaly gradient exceeding the threshold value is retained to form sequence $A'$ (only obtain data sequences of geomagnetic anomalies).

(2) The gradient of the geomagnetic anomalies that exceed the threshold on the day of an earthquake and 1–3 days after the earthquake will not be considered for prediction and will be removed.

(3) Only one gradient value of the geomagnetic anomalies continuously exceeding the threshold is retained (only one continuous multiple geomagnetic anomalies is retained).

(4) If one gradient value is selected, the corresponding value in sequence $B$ is taken 1; otherwise, 0 (1 indicates a possible earthquake, 0 indicates no earthquake).

In addition, according to the seismic data released by the China Earthquake Administration Network Center, the time of earthquake occurrence within 500 km around the monitoring area is selected. If there is an earthquake, 1 is taken, and if there is no earthquake, 0 is taken, forming the seismic time series $C$, with one datum per day.

Based on the experimental data collected in this project, it can be proven that the correlation between the predicted time sequence $B$ and the earthquake time sequence $C$ can reach over 85.7%. This proves that the sequences $B$ and $C$ has a strong correlation, which means that the probability of the geomagnetic anomaly before each earthquake is 85.7%. That is to say, it is reasonable and feasible to use geomagnetic anomaly data for earthquake prediction.

## 3. Data Analysis and Prediction Experiments

According to the data observed from the geomagnetic monitoring network and the data released by the China Earthquake Administration Network Center, a total of eight earthquakes (Table 2) occurred in and around the study area between 10 November 2021 and 6 January 2022. These earthquakes were distributed around the geomagnetic monitoring network, which the minimum magnitude was 3.0 M and the maximum magnitude was 5.5 M.

**Table 2.** Statistics of earthquakes from 10 November 2021 to 5 January 2022.

| Date | M | Location | Epicenter Distance (km) |
|---|---|---|---|
| 17 November 2021 | 3.1 | Xingwen County, Yibin City, Sichuan Province | 433 |
| 21 November 2021 | 4.6 | Changning County, Yibin City, Sichuan Province | 410 |
| 26 November 2021 | 3.3 | Weiyuan County, Neijiang City, Sichuan Province | 460 |
| 1 December 2021 | 4.2 | Changning County, Yibin City, Sichuan Province | 436 |
| 12 December 2021 | 3.0 | Yanyuan County, Liangshan Prefecture, Sichuan P. | 126 |
| 21 December 2021 | 3.3 | Ningnan County, Liangshan Prefecture, Sichuan P. | 187 |
| 2 January 2022 | 5.5 | Ninglang County, Lijiang City, Yunnan Province | 248 |
| 5 January 2022 | 3.9 | Ninglang County, Lijiang City, Yunnan Province | 246 |

The sampled magnetic intensity data are processed according to Formulas (6) and (7), and the length of time window is taken as 90 s, so the geomagnetic anomaly sequence $A$ shown in Figure 3 can be obtained. The column line sequence in Figure 3 shows the daily average gradient of the geomagnetic anomaly. It can be seen that the gradients of

geomagnetic anomalies are both positive and negative. The positive indicates that the geomagnetic anomaly becomes larger, the negative indicates that the geomagnetic anomaly becomes smaller, and the black dotted line indicates the date of the earthquake occurrence listed in Table 2. The gradient threshold of the geomagnetic anomalies is set as ±200 in Figure 3. Those exceeding the threshold are indicated in red. It can be further seen that during this period of strong earthquakes, the variation of the geomagnetic field is also very large, and the geomagnetic anomalies existed before and after the earthquake. The red magnetic anomaly value in Figure 3 is extracted to obtain sequence $A'$, sequence $B$ is obtained according to the above extraction rules, and sequence $C$ is obtained according to Table 2. Figure 4 illustrates a notable correlation between sequence $B$ and sequence $C$, with a substantial similarity of 85.7%.

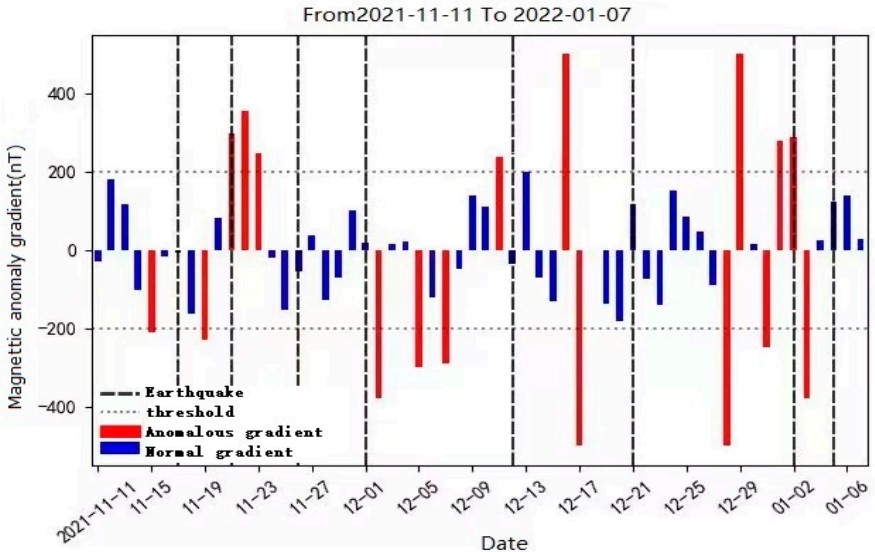

**Figure 3.** The gradients of geomagnetic anomalies from 10 November 2021 to 7 January 2022 (red columnar line).

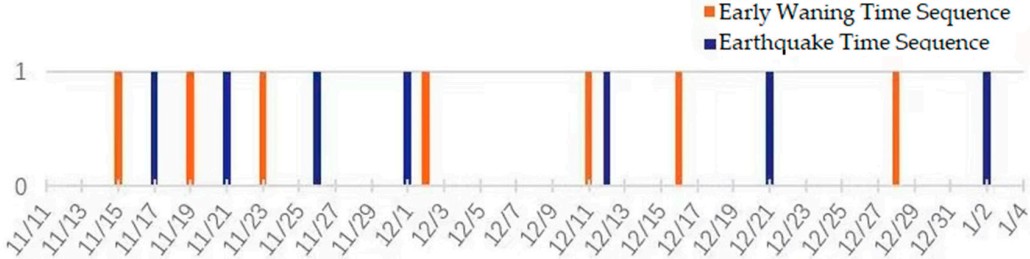

**Figure 4.** Correlation between early warning time sequence and real earthquake time sequence.

Therefore, the data in Figure 4 show that during this period (10 November 2021 to 7 January 2022), six of the seven earthquakes had a geomagnetic anomaly before, with a probability of 85.7%, so it is reasonable to predict occurrence of earthquake using the geomagnetic anomaly. Next, we try to use the geomagnetic anomaly before the earthquake to predict its occurrence.

As shown in Figure 5, the blue sequence is $A$, and according to the above extraction rules, the blue column with a red circle is sequence $B$, and the black dashed line is sequence C. Therefore, the columns with a red circle are the warning time point. It can be seen that warning signals appeared before the six earthquakes, and a total of seven predictions were made for the eight earthquakes, with six correct, one false, and one missed, resulting in an accuracy rate of 75%, a missed rate of 12.5%, and false rate of 15%. The evaluation index results are shown in Table 3. It is worth pointing out that these forecasting practices are

still too few, and the calculation of their accuracy, false alarm rate, and missed alarm rate has no statistical significance.

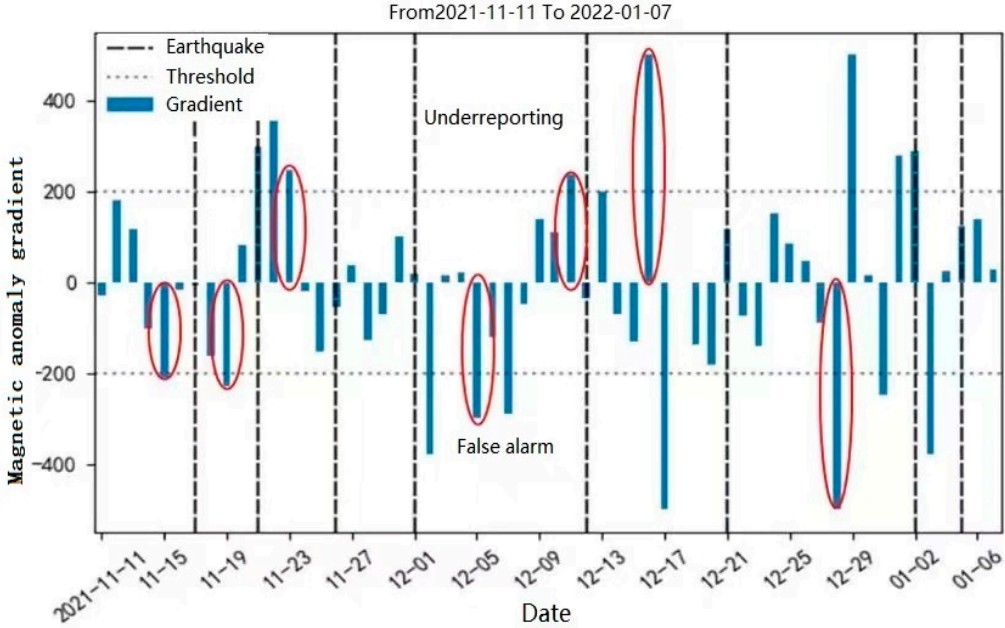

**Figure 5.** Schematic diagram of early warning points from 10 November 2021 to 7 January 2022 (the red circle is the early warning time point).

**Table 3.** The evaluation index results (10 November 2021–January 2022).

| Precision | Recall | F1 Score |
| --- | --- | --- |
| 75% | 63% | 72% |

By analyzing the above prediction results and combining them with the actual occurrence of local earthquakes, we find that because of the continuous earthquakes during this period, the interval between the two earthquakes was too close, and the geomagnetic anomaly affected each other, causing the geomagnetic anomaly to extend to about a week before the earthquake, and resulting in false and missing reports.

Let us observe another group of less dense earthquake data again: the data from 1 April 2022 to 1 June 2022. According to the data released by the China Earthquake Administration Network Center (Table 4), a total of seven earthquakes occurred during this period. The results obtained using the same data processing method and threshold are shown in Figure 6. In this figure, the blue column is sequence *A*, the column with the red circle is sequence *B*, and the black dotted line is sequence *C*. From Figure 6, we can see that seven earthquakes were predicted six times with six correct predictions, no false alarm, and one missed, so the accuracy rate is 85%, the false rate is 0, and the missed rate is 14%, the evaluation index result is shown in Table 5. The predicted time reaches 5 days before the earthquake. It is worth pointing out that both the recent earthquakes in Ya'an, Sichuan, China, were predicted 5 days in advance. For the Ya'an M4.8 earthquake on 20 May, the gradient value of geomagnetic anomaly exceeded the threshold twice on 15 and 17 May, reaching ±400 respectively. For the Ms6.1 Ya'an earthquake on 1 June, the gradient value of the geomagnetic anomaly exceeded the threshold value by 250 on 27 May.

**Table 4.** Statistics of earthquakes from 1 April 2022 to 1 June2022.

| Date | M | Location | Epicenter Distance (km) |
|---|---|---|---|
| 4 April 2022 | 3 | Qianwei County, Leshan, Sichuan Province | 353 |
| 12 April 2022 | 3 | Li County, Aba Prefecture, Sichuan Province | 450 |
| 16 April 2022 | 4.6 | Ninglang County, Lijiang City, Yunnan Province | 253 |
| 22 April 2022 | 3.2 | Kangding City, Ganzi Prefecture, Sichuan Province | 290 |
| 30 April 2022 | 4.2 | Ninglang County, Lijiang City, Yunnan Province | 251 |
| 20 May 2022 | 4.8 | Hanyuan County, Ya'an City, Sichuan Province | 252 |
| 1 June 2022 | 6.1 | Lushan County, Ya'an City, Sichuan Province | 374 |

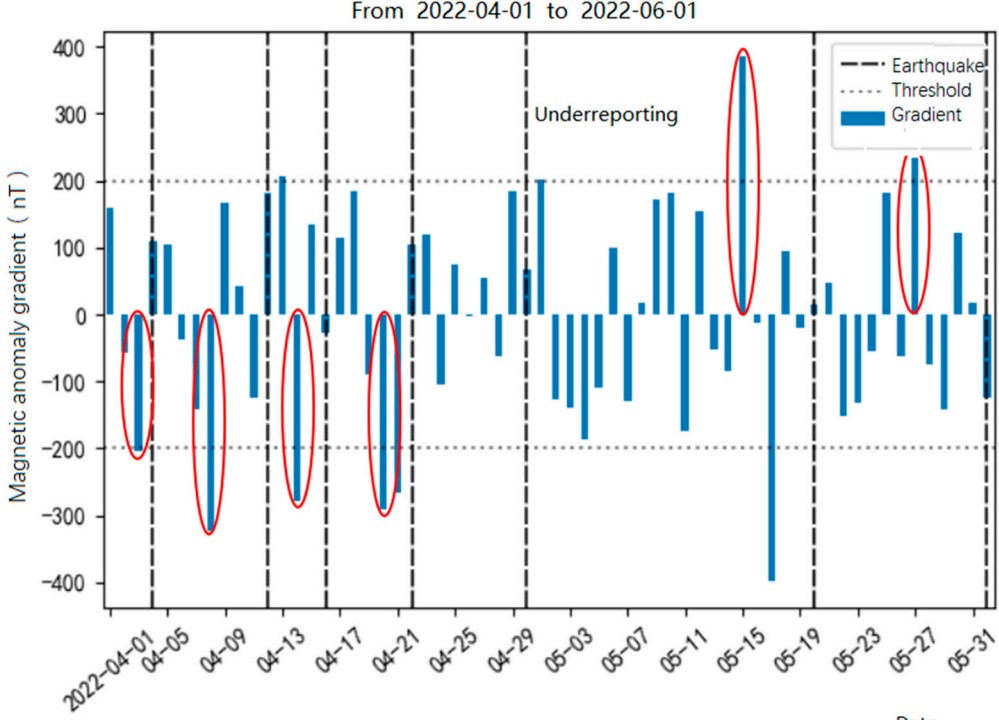

**Figure 6.** Schematic diagram of early warning points from1 April 2022 to 1 June 2022 (the red circle is the early warning time point).

**Table 5.** The evaluation index results (1 April 2022–1 June 2022).

| Precision | Recall | F1 Score |
|---|---|---|
| 85.7% | 65% | 79% |

## 4. Discussion

### 4.1. Correspondence between Geomagnetic Anomalies and Earthquakes

According to the previous literature, geomagnetic anomalies can only be observed before large earthquakes. These two sets of data proves that the magnitude of the earthquake ranges from 3.0 to 6.8, and the distance from the epicenter to the geomagnetic network center ranges from 158 km to 460 km, all of which have observed geomagnetic anomalies before the earthquakes. Based on comprehensive historical data and observations up to now, it has been shown that the majority of earthquakes undergo geomagnetic anomalies before they occur regardless of the earthquake intensity, and there is a one-to-one correspondence between geomagnetic anomalies and earthquakes. The pre-earthquake geomagnetic anomalies can serve as an important parameter for earthquake prediction.

However, the mechanism of earthquake magnetism is still unclear, and we have not yet conducted research on this issue, which requires further exploration.

### 4.2. Extraction of Impending Geomagnetic Anomalies

Based on our monitoring and data analysis, the one-to-one correspondence between geomagnetic anomalies and earthquakes depends on the strength of the gradient of geomagnetic anomalies. According to the new algorithm proposed in this article, the geomagnetic anomalies before, during, and after earthquakes can be captured, and a prediction sequence can be constructed according to the threshold and extraction rules provided in this article. The one-to-one correspondence between the predicted time sequence and the time sequence of earthquakes reaches about 75–85%.

### 4.3. The Reproducibility

After the deployment of the sensor network in 2019, using the algorithm proposed in the article, the data from 2020, 2021, 2022, and 2023 all showed the same pattern and prediction results for geomagnetic anomalies.

### 4.4. Limitations of This Method

The biggest limitation of this method is its small monitoring range. At present, monitoring can only be carried out in the area around the magnetic strength sensing network, which can mainly be used to predict the occurrence time of earthquakes within a 500 km radius of the observation area, and can provide short-term and imminent predictions within a week. It is not yet possible to provide predictions of the magnitude and epicenter of earthquakes.

### 5. Conclusions

In summary, the above results show that it is feasible to use impending geomagnetic anomalies to predict the occurrence time of earthquakes and the accuracy of using the prediction time sequence for predicting the time of an earthquake occurrence can reach 75–85%. It is worth further studying the mechanism of geomagnetic anomalies that occur with/as earthquakes happen.

And the conclusions above are only based on data analysis from Liangshan Prefecture, Sichuan Province, China in 2021 and 2022. These conclusions are correct for the area of the Liangshan Prefecture in Sichuan Province, China, but they are not necessarily universal. And it is worth further research.

**Author Contributions:** Conceptualization, P.Z.; methodology, Y.H.; software, S.L.; validation, Y.H., S.L. and P.Z.; investigation, Y.H.; writing—original draft preparation, Y.H.; writing review and editing, P.Z. All authors have read and agreed to the published version of the manuscript.

**Funding:** This research was funded by the National Natural Science Foundation of China, grant number 42074074.

**Institutional Review Board Statement:** Not applicable.

**Informed Consent Statement:** Not applicable.

**Data Availability Statement:** The data presented in this study are available in the article.

**Acknowledgments:** In this section, we would like to express our gratitude to Hao Zhang, Zhijian Peng, Bin Yang and Jiaqu Tao involved in this project.

**Conflicts of Interest:** The authors declare no conflict of interest.

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
