# Peer review of "Feasibility Study on Earthquake Prediction Based on Impending Geomagnetic Anomalies"

_applsci, doi:10.3390/app14010263_

Round 1
Reviewer 1 Report
Comments and Suggestions for Authors
While the topic is of interest, the body do not represent a novel work. The literature review is not complete and must be enriched by much more relevant work. The methodology is not clear and must be modified. The results are not well explained and shall be further discussed. The conclusion is not clear and that the limitations of the work is not well understood. Overall, the work cannot be recommended for publication here.
Author Response
Response: We have carefully revised the manuscript to address various issues pointed out by the reviewers.
Reviewer 2 Report
Comments and Suggestions for Authors
The study presents intriguing findings demonstrating detectable signatures on magnetic sensors during earthquake events. While the paper makes a valuable contribution, there are several areas where the manuscript's quality can be enhanced. The following comments and suggestions aim to address these issues:
-
The manuscript's introduction suggests the possibility of an earthquake warning system based on magnetic sensors, although there doesn't appear to be a predictive model. Clarify this point to avoid potential confusion.
-
In the methodology section, the definition of the stochastic process (mentioned between lines 71-77) should be formalized, including a definition of mean square deviation. Consider specifying whether f(t) is a Gaussian process.
-
Provide a table with detailed specifications for the model AM2-1001 sensor.
-
In the description between lines 82-85, include an appendix with station coordinates for clarity.
-
Address the low resolution of Figure 2 and the inadequate quality of the map in Figure 1. Ensure that the red circles in Figure 1 are clearly discernible and accurately represented. Improve the captions of the figures to enhance their descriptions.
-
Clearly define the date of the earthquake event used in the study. For example, in Figure 2, provide the event's date to give readers context.
-
In line 89, clarify that the mentioned stations (circled in red) exhibited atypical behavior during the 2019 event, but this behavior may not be consistent across all events. Explain why not all stations behaved similarly, if possible.
-
In line 102, specify the magnitude of "weak" to provide a clearer understanding.
-
Ensure that all methods referenced in the manuscript are properly cited, including the one mentioned in line 125.
-
The description in section 2.3 could benefit from more detail to improve clarity and comprehensibility.
-
Restructure the final section containing the discussion and conclusion to align with the typical logic of scientific texts. Ensure that it flows logically and effectively summarizes the key findings and implications.
Author Response
The study presents intriguing findings demonstrating detectable signatures on magnetic sensors during earthquake events. While the paper makes a valuable contribution, there are several areas where the manuscript's quality can be enhanced. The following comments and suggestions aim to address these issues:
- The manuscript's introduction suggests the possibility of an earthquake warning system based on magnetic sensors, although there doesn't appear to be a predictive model. Clarify this point to avoid potential confusion.
Response: The algorithm and extraction rules proposed in the paper are prediction algorithms, which can also be called the prediction model in mathematics.
- In the methodology section, the definition of the stochastic process (mentioned between lines 71-77) should be formalized, including a definition of mean square deviation. Consider specifying whether f(t) is a Gaussian process.
Response: We assume that a time series of geomagnetic observation, , as an ergodic stationary stochastic process. Its mean can be expressed as: , and its standard deviation is S. Magnetic anomaly is defined as: when , ΔF is called magnetic anomaly.
- Provide a table with detailed specifications for the model AM2-1001 sensor
|
Specifications of AM2-1001 sensor |
|
|
Meas. range |
±100 µT, other ranges on request |
|
Accuracy at 20 â—¦C |
±2% ± 0.3 µT |
|
Operating temperature |
−40 to +85 â—¦C |
|
Zero drift |
<2 nT/K |
|
Output voltage OUT+ ref. to OUT− |
±1 V/50 µT, max. ±2,5 V |
|
Bandwidth |
0 to 1 kHz (−3 dB) |
|
DC output impedance |
<1 Ω |
|
Reference output OUT− |
2.5 V ref. to supply ground (0 V) |
|
Max. load between OUT+ and OUT− |
>1 kΩ, <100 pF |
|
Noise |
<0.5 nTRMS or 3 nTPP (0.1 to 10 Hz) ,∼150 pT/√Hz @ 1 Hz |
|
Supply voltage |
5 V ±5% |
|
Supply current |
∼2 mA |
|
Dimensions |
44.5 mm × 14 mm × 5.5 mm |
|
Length of detection coil |
22mm |
- In the description between lines 82-85, include an appendix with station coordinates for clarity.
Response: The magnetic sensing network is deployed in Liangshan Prefecture, China, with a longitude of 101.71°E~102.62°E and a latitude of 27.29°N~28.92°N. Due to the large number of stations, each point is not listed one by one.
- Address the low resolution of Figure 2 and the inadequate quality of the map in Figure 1. Ensure that the red circles in Figure 1 are clearly discernible and accurately represented. Improve the captions of the figures to enhance their descriptions.
Response: This diagram has been redrawn.
- Clearly define the date of the earthquake event used in the study. For example, in Figure 2, provide the event's date to give readers context.
Response: It has been modified in Figure 2.
- In line 89, clarify that the mentioned stations (circled in red) exhibited atypical behavior during the 2019 event, but this behavior may not be consistent across all events. Explain why not all stations behaved similarly, if possible.
Response: The magnetic field intensity of each station in the network not only depends on the distance of the station from the epicenter, but also is related to the underground geological structure and rock magnetism. This is a challenging issue that requires further study and investigation.
- In line 102, specify the magnitude of "weak" to provide a clearer understanding.
Response: Generally speaking, the smaller the magnitude of the earthquake, the relatively "weaker" the geomagnetic anomaly, where weak means "small".
- Ensure that all methods referenced in the manuscript are properly cited, including the one mentioned in line 125.
Response: We have checked all of the references.
- The description in section 2.3 could benefit from more detail to improve clarity and comprehensibility.
Response: We provided explanations in parentheses for rules (1) to (4) in section 2.3
- Restructure the final section containing the discussion and conclusion to align with the typical logic of scientific texts. Ensure that it flows logically and effectively summarizes the key findings and implications.
Response: We have made revisions in accordance with the reviewer's comments. See the discussion and conclusions section of the new version of the manuscript for details.

Reviewer 3 Report
Comments and Suggestions for Authors
The manuscript is of great interest for evaluating the possibility of predicting earthquakes. I have no doubt that the content of the paper is in line with the subject of the journal. However, in order for the manuscript to be published, the authors should pay attention to the following observations.
1. In order to draw conclusions based on the analysis of magnetic anomalies, it is necessary to give a more complete description of the Earth's magnetic field, its components, especially those that are directly or indirectly related to seismic activity.
2. In the introduction it is necessary to give a more complete qualitative and quantitative analysis of the relationship between earthquakes and magnetic field variations recorded before them. Present modern data for different regions of the world.
3. The list of references is very incomplete.
4. Line 34: "Super low frequency (ULF) range" - ULF is Ultra Low Frequency. Before using an abbreviation (e.g. VAN) for the first time, the full name of the method should be given.
5. Magnetic anomalies have different nature, intensity, frequency, time of manifestation, etc. The authors declare the development of an algorithm for predicting earthquakes. This is a very interesting topic. However, in the manuscript it requires a more detailed description. It is not clear how to separate magnetic field variations associated with the occurrence of seismic activity from anomalies due to other geodynamic or cosmic causes.
6. Of particular interest is the authors' report in the discussion section that magnetic anomalies can be indicators of weak earthquakes. However, this issue is practically not covered.
7. There are practically no conclusions in the article.
Comments on the Quality of English LanguageAuthors need to proofread the text carefully, correct inaccurate abbreviation.
Author Response
The manuscript is of great interest for evaluating the possibility of predicting earthquakes. I have no doubt that the content of the paper is in line with the subject of the journal. However, in order for the manuscript to be published, the authors should pay attention to the following observations.
- In order to draw conclusions based on the analysis of magnetic anomalies, it is necessary to give a more complete description of the Earth's magnetic field, its components, especially those that are directly or indirectly related to seismic activity.
Response: Based on the reviewer’s comment, we have added the following to the new manuscript:
The Earth's global magnetic field is composed of the normal field generated by the movement of matter within the Earth, as well as the diurnal and annual changing magnetic fields caused by the sun. Additionally, certain ore bodies and special rocks found within the Earth, such as basalt, generate additional magnetic fields known as local magnetic fields. Seismic activity, which involves the movement of matter, can also produce local magnetic fields known as geomagnetic anomalies. These anomalies have been confirmed through numerous observations. However, the exact mechanism by which seismic activity generates geomagnetic anomalies remains unclear, presenting a challenging scientific issue.
- In the introduction it is necessary to give a more complete qualitative and quantitative analysis of the relationship between earthquakes and magnetic field variations recorded before them. Present modern data for different regions of the world.
Response:
In this study, we have conducted extensive statistical and analytical work on the geomagnetic field in our study area. The earthquake prediction presented in this article is based on the findings derived from this work. However, due to the constraints of the article's focus and length, only a limited number of geomagnetic field analysis results are presented in Sections 2 and 3. Furthermore, it is important to note that our study area is confined to the Liangshan area in Sichuan Province, China. Unfortunately, other regions do not have a comparable density of geomagnetic observation networks, making it impractical to investigate geomagnetic fields in those areas or on a global scale.
- The list of references is very incomplete.
Response: We have complete the list of references.
- Line 34: "Super low frequency (ULF) range" - ULF is Ultra Low Frequency. Before using an abbreviation (e.g. VAN) for the first time, the full name of the method should be given.
Response:
VAN is the abbreviation of three Greek scientist’s names, P. Varotsos, K. Alexopoulos and K. Nomicos. Provide a brief introduction to the VAN method and provide references in my manuscript 。
- Magnetic anomalies have different nature, intensity, frequency, time of manifestation, etc. The authors declare the development of an algorithm for predicting earthquakes. This is a very interesting topic. However, in the manuscript, it requires a more detailed description. It is not clear how to separate magnetic field variations associated with the occurrence of seismic activity from anomalies due to other geodynamic or cosmic causes.
Response:
In our earthquake prediction, we only consider the intensity of geomagnetic anomalies. Additionally, geomagnetic anomalies can also include solar magnetic storms. However, when solar magnetic storms are observed, the anomalous data associated with them are not taken into account in earthquake prediction.
- Of particular interest is the authors' report in the discussion section that magnetic anomalies can be indicators of weak earthquakes. However, this issue is practically not covered.
Response:
The focus of this article is the discovery that earthquakes of magnitude 3-4 can also produce geomagnetic anomalies, which have been successfully applied in earthquake prediction. The content of this article is primarily concerned with the prediction of minor earthquakes, as our study area has only experienced one significant earthquake (magnitude 6.8).
- There are practically no conclusions in the article.
Response: The key conclusion is that both small and large earthquakes exhibit pre-seismic geomagnetic anomalies, indicating that these anomalies can serve as geomagnetic precursors. Importantly, the new algorithm proposed in this paper can effectively capture pre-seismic magnetic anomalies, although not always successfully. By following the extraction rules provided in this paper, it is possible to construct prediction sequences and make pre-seismic forecasts, thus making it feasible to utilize geomagnetic anomalies for earthquake prediction.

Reviewer 4 Report
Comments and Suggestions for Authors
REVISION MANUSCRIPT Applied Sciences- 2574941: “Feasibility study on earthquake prediction based on impending Geomagnetic anomalies.
Specific comments:
Comment 1: Data and sample size: The paper does not provide details on the size and diversity of the dataset used for analysis. A larger and more diverse dataset would enhance the robustness and generalizability of the findings.
Comment 2: Methodology clarity: while the paper mentions the proposed algorithm for predicting earthquakes based on geomagnetic anomalies, it lacks a detailed explanation of the algorithm’s mechanics. A step-by-step breakdown of how the algorithm works would aid in understanding and replicating the research.
Comment 3: External interference handling: The paper mentions the accumulation of geomagnetic anomaly energy to eliminate external environmental interference. However, it does not elaborate on the specific techniques or methodologies used for this purpose. Providing more details on how external interference is managed would strengthen the methodology.
Comment 4: Statistical analysis: The paper cites an accuracy range of 75% to 85% for earthquake prediction, but it does not delve into the statistical methods or metrics used to evaluate these predictions. A thorough explanation of the evaluation metrics, such as precision, recall, or F1 score, would provide a clearer picture of the predictive performance
Comment 5: Correlation vs causation: While the paper establishes a correlation between geomagnetic anomalies and earthquake occurrence, it does not conclusively prove causation. Further investigation into the physical mechanisms underlying this relationship would enhance the papers scientific rigor.
Comment 6: Validation and reproducibility: The paper would benefit from discussing how the results were validated and whether the findings have been reproduced independently.
Comment 7: Discussion of limitations: The paper briefly mentions the relationship between geomagnetic anomalies and earthquake characteristics but does not delve deeply into this aspect. A more extensive discussion of the limitations of the study, such as the range of earthquake magnitudes and depths considered, would provide a more comprehensive view of the research scope.
Comment 8: Practical Application: While the paper suggests that geomagnetic anomalies could be used for earthquake prediction, it does not address the practical challenges and feasibility of implementing such a system in real-world scenarios. Considerations like cost, infrastructure requirements, and false positive/negative rates should be discussed.
Comment 9: Data availability: If possible, the paper could mention whether the geomagnetic data used in the study is publicly available for other researchers to verify and build upon the findings.
In summary, while the paper presents intriguing findings regarding the correlation between geomagnetic anomalies and earthquake occurrence, it could benefit from providing more in-depth methodological details, addressing limitations, and discussing the practical implications and challenges of using this approach in earthquake prediction systems.
Comments on the Quality of English LanguageRegarding the English used in the paper, there are a few areas where improvement is needed:
While the paper conveys its main points, there are instances where sentence structure and paragraph organization could be clearer for the reader.
There are minor grammatical errors and punctuation issues throughout the text.
Maintain consistency in terminology, especially when referring to variables or concepts. Ensure that the same terminology is used throughout the paper to avoid confusion.
Some sentences in the paper are quite long and complex, which can make them challenging to follow. Consider breaking long sentences into shorter.
Author Response
Comment 1: Data and sample size: The paper does not provide details on the size and diversity of the dataset used for analysis. A larger and more diverse dataset would enhance the robustness and generalizability of the findings.
Response: A total of 30 instruments were deployed in the geomagnetic network, with each magnetometer recording the three components of the geomagnetic field (x, y, and z) every 3 seconds. Starting from September 2019, we have obtained approximately four years of monitoring data, amounting to around 100 KB of data per day. As of now, we have accumulated a total of approximately 1.25 GB of data.
Comment 2: Methodology clarity: while the paper mentions the proposed algorithm for predicting earthquakes based on geomagnetic anomalies, it lacks a detailed explanation of the algorithm’s mechanics. A step-by-step breakdown of how the algorithm works would aid in understanding and replicating the research.
Response: More details have been added to sections 2 and 3 of the manuscript.
Comment 3: External interference handling: The paper mentions the accumulation of geomagnetic anomaly energy to eliminate external environmental interference. However, it does not elaborate on the specific techniques or methodologies used for this purpose. Providing more details on how external interference is managed would strengthen the methodology.
Response: Mathematical proof have been added to section 2.2 of the manuscript.
.
Comment 4: Statistical analysis: The paper cites an accuracy range of 75% to 85% for earthquake prediction, but it does not delve into the statistical methods or metrics used to evaluate these predictions. A thorough explanation of the evaluation metrics, such as precision, recall, or F1 score, would provide a clearer picture of the predictive performance
Response: We have added Tab. 2. The evaluation index results(2021-11-10-2022-1-7) and Tab. 4. The evaluation index results (2022-04-01 to 2022-06-01) in section 3.
Comment 5: Correlation vs causation: While the paper establishes a correlation between geomagnetic anomalies and earthquake occurrence, it does not conclusively prove causation. Further investigation into the physical mechanisms underlying this relationship would enhance the papers scientific rigor.
Response: The mechanism by which seismic activity generates geomagnetic anomalies remains unclear and poses a challenging scientific problem. This is a subject that requires a significant number of indoor and field experiments, and it is quite difficult. We are currently undertaking research in this area.
Comment 6: Validation and reproducibility: The paper would benefit from discussing how the results were validated and whether the findings have been reproduced independently.
Response: After deploying the geomagnetic network in 2019, using the algorithm proposed in the paper, the geomagnetic data related to earthquakes in 2020, 2021, 2022, and 2023 all exhibited nearly identical characteristics and patterns.
Comment 7: Discussion of limitations: The paper briefly mentions the relationship between geomagnetic anomalies and earthquake characteristics but does not delve deeply into this aspect. A more extensive discussion of the limitations of the study, such as the range of earthquake magnitudes and depths considered, would provide a more comprehensive view of the research scope.
Response: We have revised the paper and mentioned it in the conclusion.
Comment 8: Practical Application: While the paper suggests that geomagnetic anomalies could be used for earthquake prediction, it does not address the practical challenges and feasibility of implementing such a system in real-world scenarios. Considerations like cost, infrastructure requirements, and false positive/negative rates should be discussed.
Response: This article only theoretically and technically demonstrates the feasibility of using geomagnetic anomalies to predict earthquakes. The measured data further confirms the correctness of the method. It does not cover the cost, infrastructure requirements, and other potential issues that may arise in earthquake prediction. Subsequent research will address the questions raised by the reviewers.
Comment 9: Data availability: If possible, the paper could mention whether the geomagnetic data used in the study is publicly available for other researchers to verify and build upon the findings.
Response: Scholars interested in the data of this article can contact us via email to obtain the data.
In summary, while the paper presents intriguing findings regarding the correlation between geomagnetic anomalies and earthquake occurrence, it could benefit from providing more in-depth methodological details, addressing limitations, and discussing the practical implications and challenges of using this approach in earthquake prediction systems.
Comments on the Quality of English Language
Regarding the English used in the paper, there are a few areas where improvement is needed:
While the paper conveys its main points, there are instances where sentence structure and paragraph organization could be clearer for the reader.
There are minor grammatical errors and punctuation issues throughout the text.
Maintain consistency in terminology, especially when referring to variables or concepts. Ensure that the same terminology is used throughout the paper to avoid confusion.
Some sentences in the paper are quite long and complex, which can make them challenging to follow. Consider breaking long sentences into shorter.
Response: We have carefully revised the manuscript in response to the language issues pointed out by the reviewers, hoping to meet the requirements for publication.

Reviewer 5 Report
Comments and Suggestions for Authors
The novelty of the manuscript is not suitable for publication. Further, very limited literatures and contents are mentioned in the manuscript. Results and analysis are also less. Presentation of the paper also needs major improvement. It is recommended not to use I/we/he in the manuscript.
Comments on the Quality of English LanguageModerate editing of English language required
Author Response

(The authors gave the same response as above.)

Round 2
Reviewer 2 Report
Comments and Suggestions for Authors
The work is interesting and contains information that is worth publishing in general. However, section 4 is below the standards for an Applied Sciences paper. A section with 16 lines and 4 subsections is not acceptable to me. I had already asked for this in the previous round of review and I will again recommend that this section be reworked by calling it a conclusion and with the findings of the work properly addressed.
Author Response
Dear Sri,
Thanks for your comments and suggestions!
In response to your question,we have made modifications.See in Line 297 -318 in blue font in the paper.
Look for your suggestions

Reviewer 4 Report
Comments and Suggestions for Authors
REVISION MANUSCRIPT Applied Sciences- 2574941: “Feasibility study on earthquake prediction based on impending Geomagnetic anomalies.”
Reviewer #:
General Comments:
Thank you very much for considering me to review this manuscript again! I see that the authors took into account all my previous observations!
The authors worked so hard and well on the manuscript revisions and now I think their paper is suitable for publication!
Author Response
Dear sir,
Thanks for your Comments! Minor editing of English language has revised.
Reviewer 5 Report
Comments and Suggestions for Authors
Accept in present form
Comments on the Quality of English LanguageMinor editing of English language required
Author Response
Dear Sir,
Thanks for your Accept!
Minor editing of English language has revised.See in Line 256-259,and Line 278-280.
Some conclusions has revised.See in Line 297-318
Look for your suggestion.
Round 3
Reviewer 2 Report
Comments and Suggestions for Authors
The paper presented substantial improvements overall. I still don't agree with the last section being called 'discussion and conclusion.' In my opinion, discussions should be moved to section 3, and the last section should focus only on a section that needs improvement.
Author Response
Dear Expert Sir,
We separated the discussion from the conclusion. This really makes the logic much clearer. Thank you for your suggestion.
